# Environmental Enrichment Engages Vesicular Zinc Signaling to Enhance Hippocampal Neurogenesis

**DOI:** 10.3390/cells12060883

**Published:** 2023-03-13

**Authors:** Michael J. Chrusch, Selena Fu, Simon C. Spanswick, Haley A. Vecchiarelli, Payal P. Patel, Matthew N. Hill, Richard H. Dyck

**Affiliations:** 1Hotchkiss Brain Institute, University of Calgary, Calgary, AB T2N 1N4, Canada; mikechrusch@gmail.com (M.J.C.); selena.fu2@ucalgary.ca (S.F.); scspansw@ucalgary.ca (S.C.S.); haleyvecchiarelli@uvic.ca (H.A.V.); mnhill@ucalgary.ca (M.N.H.); 2Department of Neuroscience, University of Calgary, Calgary, AB T2N 1N4, Canada; 3Department of Psychology, University of Calgary, Calgary, AB T2N 1N4, Canada; 4Department of Cell Biology and Anatomy, University of Calgary, Calgary, AB T2N 1N4, Canada

**Keywords:** neurogenesis, hippocampus, adult hippocampal neurogenesis, experience-dependent plasticity, zinc, zinc transporter 3

## Abstract

Zinc is highly concentrated in synaptic vesicles throughout the mammalian telencephalon and, in particular, the hippocampal dentate gyrus. A role for zinc in modulating synaptic plasticity has been inferred, but whether zinc has a particular role in experience-dependent plasticity has yet to be determined. The aim of the current study was to determine whether vesicular zinc is important for modulating adult hippocampal neurogenesis in an experience-dependent manner and, consequently, hippocampal-dependent behaviour. We assessed the role of vesicular zinc in modulating hippocampal neurogenesis and behaviour by comparing ZnT3 knockout (KO) mice, which lack vesicular zinc, to wild-type (WT) littermates exposed to either standard housing conditions (SH) or an enriched environment (EE). We found that vesicular zinc is necessary for a cascade of changes in hippocampal plasticity following EE, such as increases in hippocampal neurogenesis and elevations in mature brain-derived neurotrophic factor (mBDNF), but was otherwise dispensable under SH conditions. Using the Spatial Object Recognition task and the Morris Water task we show that, unlike WT mice, ZnT3 KO mice showed no improvements in spatial memory following EE. These experiments demonstrate that vesicular zinc is essential for the enhancement of adult hippocampal neurogenesis and behaviour following enrichment, supporting a role for zincergic neurons in contributing to experience-dependent plasticity in the hippocampus.

## 1. Introduction

Zinc is sequestered into the synaptic vesicles of a subset of glutamatergic neurons in the mammalian forebrain [1,2]. In zincergic neurons, zinc is transported into synaptic vesicles by zinc transporter 3 (ZnT3, *SLC30A3*) [3] and released into the synapse in an activity-dependent manner [4,5]. In the synapse, zinc has the potential to activate or modulate a number of signaling pathways by binding to specific sites on NMDA receptors [6] or by transactivating TrkB receptors [7], among others (reviewed in [2,5]). Zinc is conspicuously contained, at especially high levels, in the synaptic terminals of the granule cells of the dentate gyrus [8], providing the potential for zinc signaling to potently affect hippocampal function. However, early research assessing the role of synaptic zinc in hippocampal function by assessing mice devoid of vesicular zinc through the genetic ablation of ZnT3 [9] revealed little functional or behavioural consequences [10,11]. Recent research has revealed that hippocampal deficits in ZnT3 KO mice emerge with age. While 3-month-old ZnT3 KO mice had no detectable deficits, spatial memory deficits were found in 6-month-old ZnT3 KO mice, and these deficits were related to decreased hippocampal protein levels of the neural precursor marker doublecortin, as well as decreased levels of pro-brain-derived neurotrophic factor (proBDNF) and TrkB protein [12]. This would suggest that while vesicular zinc may be dispensable for normative hippocampal function, there may be specific ontogenetic or experiential conditions in which the importance of zincergic signaling becomes apparent.

With respect to contributing to hippocampal plasticity, it is important to note that a number of studies have shown that zinc is a potent modulator of brain-derived neurotrophic factor (BDNF) signaling, capable of directly transactivating the TrkB receptor as well as promoting the activation of matrix metalloproteinases (MMPs) which act in the conversion of proBDNF to mature BDNF (mBDNF) [7,13]. mBDNF signaling via the TrkB receptor is particularly important in the proliferation, differentiation, and survival of newborn neurons [14,15]. In addition to its effects on neurotrophic factors, zinc, as an essential mineral, is critical for the proliferation and survival of cells in the adult hippocampus [16]. However, the specific role(s) of synaptically released zinc are largely unknown. The continuous generation of new neurons throughout adulthood is a defining feature of the hippocampus, and integration of these neurons into hippocampal circuitry plays an important role in hippocampal-dependent learning and memory [17], but its precise roles and functions are still being debated. The magnitude of hippocampal neurogenesis can be modulated by experience, and most studies evaluating the effects of exercise and exposure to an enriched environment (EE) support the idea that the addition of newborn adult neurons may be beneficial for learning and memory abilities. For example, voluntary running and exposure to an EE increase hippocampal neurogenesis, which is paralleled by improved spatial memory performance in the Morris Water task [18,19,20]. This increase in neurogenesis is accompanied and, perhaps, mediated by increased levels of BDNF [21,22]. Newborn granule cells have also been shown to preferentially integrate into circuits supporting spatial memory [23]. More recent work has demonstrated that adult hippocampal neurogenesis mediates pattern separation, with increased neurogenesis improving the ability to discriminate between nearby stimulus locations on a touchscreen task [24] and between two similar contextual-fear contexts [25]. However, studies investigating the effects of decreasing neurogenesis have produced inconsistent results. For instance, several studies have found that reducing neurogenesis has no effect on spatial navigation learning in the Morris Water task [26,27,28], but does impair long-term spatial memory in the Morris Water task [27,28]. Hindering or ablating neurogenesis has also been reported to impair contextual fear conditioning [29,30], while other studies have not shown the same thing [26,31]. It is important to note that adult hippocampal neurogenesis can be regulated at the levels of cell proliferation, differentiation, and survival. For instance, while both EE and exercise improve spatial learning ability, these two manipulations are dissociable in that EE increases the survival of new cells, whereas voluntary exercise increases the level of cell proliferation [32]. Thus, despite efforts to decode the functional role of adult neurogenesis, the precise role it plays in cognition is still unclear.

Given the anatomical localisation and integration with factors regulating plasticity, we sought to determine if, despite its lack of effect on steady-state hippocampal structure and function, vesicular zinc is involved in experience-dependent changes. Intriguingly, our data support previous findings of a lack of effect of ablation of vesicular zinc on basal hippocampal neurogenesis, mBDNF levels, and hippocampal-dependent behaviour, but demonstrate a novel role for vesicular zinc in regulating experience-dependent changes in all of these factors.

## 2. Materials and Methods

### 2.1. Animals

ZnT3 KO and WT breeding stock, generated on a C57BL/6 × 129/Sv hybrid genetic background, were generously provided by Dr. Richard Palmiter (University of Washington, Seattle, WA, USA) and utilised in this study. All animals were provided a standard laboratory diet (PicoLab Irradiated Rodent Diet 20; #5058; LabDiet, St. Louis, MO, USA) and water ad libitum and were kept under a 12:12 h light/dark cycle (lights on during the day) for the duration of the experiment. All procedures were approved by the Life and Environmental Sciences Animal Care Committee at the University of Calgary (Study ID: #BI08R-02) and conformed to the guidelines set out by the Canadian Council on Animal Care.

### 2.2. Housing Manipulation

Male ZnT3 KO and WT mice were raised in standard laboratory housing (SH) until postnatal day 60 (P60). The SH environment consisted of a transparent plastic cage (28 × 17.5 × 12 cm) containing sawdust bedding and a small plastic mouse house (Otto Environmental, Greenfield, WI, USA). At P60, ZnT3 KO and WT mice were randomly assigned to SH or enriched environment (EE) conditions. EE housing consisted of a 60 × 60 × 60 cm bi-level cage containing ladders, a variety of toys, wire mesh, tunnels, and a running wheel. Three to four mice were placed in each SH cage, while 6–8 mice were placed in the EE cage.

### 2.3. BrdU Procedure

To label actively dividing cells, mice received three intraperitoneal injections of the thymidine analog, 5′-bromo-2′-deoxyuridine (BrdU; 100 mg/kg; Sigma-Aldrich, Oakville, ON, Canada), dissolved in 0.1 M phosphate buffered saline (PBS), 6 h apart on P81. The dosage and number of injections used ensure that a sufficient number of cells are labelled with BrdU for quantitative analyses without inducing toxic effects in animals [33]. ZnT3 KO and WT mice from either SH or EE conditions were randomly assigned to proliferation (*n* = 24; 6 mice per group; killed 24 h after the final BrdU injection) or survival (*n* = 23; 5–6 mice per group; killed 3 weeks after the final BrdU injection). A 3-week period was chosen to assess cell survival, as new cells will begin to express mature neuronal proteins and thus can be phenotyped using endogenous markers such as NeuN [34,35,36].

### 2.4. Tissue Preparation and Labeling

Mice were killed with an overdose of sodium pentobarbital (400 mg/kg) and transcardially perfused with 0.1 M PBS, followed by 4% paraformaldehyde (PFA) in 0.1 M PBS. Brains were removed, post-fixed in 4% PFA at 4 °C for 24 h, and cryoprotected in 0.1 M PBS containing 30% sucrose and 0.02% sodium azide. Brains were cut into six series of 40μm coronal sections using a freezing, sliding microtome (American Optical, Model #860; Buffalo, NY, USA).

To identify BrdU-positive cells, sections from a single series were placed in 2N HCl for 60 min to denature the DNA and then washed in 6 rinses of 0.1 M PBS over a period of 90 min. Sections were then incubated in a primary antibody solution containing rat anti-BrdU (1:200, AbD Serotec MCA2060, Mississauga, ON, Canada), mouse anti-NeuN antibody (1:2000, Millipore Sigma-Aldrich, Oakville, ON, Canada), 0.3% Triton-X, and 2% normal goat serum in 0.1 M PBS for 24 h at 4 °C. Sections were washed three times for 7 min each and then incubated in a secondary solution containing goat anti-rat biotin (1:500, Jackson ImmunoResearch Laboratories 112-065-167, West Grove, PA, USA) and goat anti-mouse Cy2 (1:500, Jackson ImmunoResearch Laboratories 115-225-166, West Grove, PA, USA) in 0.1 M PBS for 24 h at 4 °C. Following three 7 min washes in 0.1 M PBS, tissue sections were incubated in 0.1 M PBS containing the tertiary antibody Alexa Fluor Streptavidin 594 (1:500, Molecular Probes, Eugene, OR, USA) for 1 h. All sections were counterstained with 4′,6′-diamidino-2-phenylindole (DAPI; 1:1000, Sigma-Aldrich, Oakville, ON, Canada) added for the final 20 min, followed by three 7 min washes in 0.1 M PBS. Sections were mounted on gelatin-coated slides, coverslipped with fluorescent mounting media, and stored in the dark at 4 °C until quantification.

To assess cell death and hippocampal granule cell layer volume, an additional series of tissue sections from the survival cohort was stained with cresyl violet. Our protocol was as follows: 1 min in 70% EtOH, 5 min in 95% EtOH, 10 min in 100% EtOH, 20 min in Xylene, 5 min in 100% EtOH, 1 min in 95% EtOH, 1 min in 50% EtOH, and 0.5 min in distilled water. Sections were then stained with 0.1% cresyl violet solution (pH 3.5, with acetic acid) for 2 min. Excess stain was removed by briefly dipping the sections in distilled water, followed by 10 min in 100% EtOH and 5 min in 100% EtOH. Sections were transferred to xylene for 10 min prior to being coverslipped with Permount^TM^ Mounting Medium (Thermo Fisher Scientific, Eugene, OR, USA). Slides were stored flat and allowed to dry prior to analysis.

### 2.5. Microscopy and Cell Quantification

To determine the extent of cell proliferation and survival in ZnT3 KO and WT mice, the total number of BrdU-positive cells within the granule cell layer and subgranular zone across the entire dentate gyrus was counted. To minimise edge artifacts, a guard zone was established in the uppermost focal plane, and only BrdU-positive cells beneath the focal plane were counted [37]. All cell counts were performed with a 63X/1.4NA objective mounted on a Zeiss Axioscope 2 microscope.

To determine the phenotype of surviving cells, the co-localization of BrdU and NeuN was assessed in 25–35 randomly selected cells spanning the rostro-caudal extent of the dentate gyrus. Using a Nikon C1s confocal microscope with a 60X/1.3NA objective, BrdU/NeuN co-localization was confirmed using z-stacks taken throughout the entirety of each selected BrdU-positive cell. A cell was considered positive for BrdU/NeuN if a nuclear signal from both the 488 nm and 561 nm lasers colocalised in the x-, y-, and z-axes [38,39].

Apoptotic and necrotic cells are defined by the condensation of chromatin within the nucleus (pyknosis) [40]. Pyknotic nuclei can be counted in cresyl violet-stained sections to effectively assess the prevalence of cell death [38,41]. Pyknotic nuclei were counted throughout the granule cell layer and subgranular zone using a 100X/1.3 objective.

The volume of the dentate gyrus granule cell layer was calculated using Cavalieri’s principle [42]. Images of cresyl violet-stained sections were quantified using Image J [43] as described previously in Spanswick and Dyck [44].

### 2.6. Behavioural Analysis

For behavioural studies, SH and EE mice (*n* = 39; 9–10 mice per group) underwent behavioural testing three weeks after SH or EE exposure. Prior to behavioural testing, all mice were individually handled for five, 2 min long sessions per day for two consecutive days in order to habituate them to handling.

#### 2.6.1. Spatial Object Recognition

The Spatial Object Recognition task consisted of three phases: the habituation, sampling, and testing phases. During the habituation phase, mice were placed in a white square box (40 × 40 × 40 cm), devoid of any objects, for 15 min on two consecutive days. The third day consisted of the sample and testing phases. During the sample phase, two identical objects were placed in the northeast (Position A) and northwest (Position B) corners of the box, and mice were given 5 min to explore both objects. After 5 min, the mouse was removed from the box for a rest period of 30 min. During the testing phase, one of the objects was moved to a novel, southern location (Position C). The mice were then returned to the box and allowed 5 min to explore. The time spent investigating the novel location (object in Position C) divided by the total investigation time for both objects during the testing phase, multiplied by 100, was calculated to determine the novel investigation ratio (%). Investigation was defined as a subject directing the nose towards the object within a 2 cm distance from the nose to the object.

#### 2.6.2. Morris Water Task

An accelerated version of the Morris Water task was adapted from Stone et al. [45]. The Morris Water task equipment consisted of a white circular pool measuring 123.5 cm in diameter with 36 cm high walls and a transluscent square platform (12 × 12 cm) submerged 1.5 cm below the surface of the water. The pool was filled with water (21 °C) to a depth of 22 cm. For each of the three days, mice were trained on four trials. For each trial, the mouse was released from one of four starting locations, the order of which was pseudo-randomly determined. Once the mouse located the hidden platform, it remained there for 15 s before being removed from the pool and placed back in its home cage for the inter-trial interval (~5–10 min). If the platform was not located within 60 s, the mouse was manually guided to the platform, and a maximum latency score of 60 s was assigned. For each trial during the training period, latency to find the platform, path length, and swimming speed were tracked and measured with an overhead camera and Ethovision XT tracking software (Noldus, Wageningen, The Netherlands).

### 2.7. BDNF Analysis and Zinc Incubated Slices

A third cohort of SH and EE mice (*n* = 32; 7–9 mice per group) were killed three weeks into SH or EE placement to determine the effects of enrichment on hippocampal levels of mature BDNF (mBDNF). ZnT3 KO and WT mice housed in either SH or EE were anaesthetised using isoflurane and killed by decapitation. The hippocampi were rapidly dissected, frozen, and stored in microcentrifuge tubes at −80 °C. To examine the acute effects of zinc application to hippocampal tissue, sagittal slices containing the hippocampus were prepared from 2-month-old mice (*n* = 29; 7–9 mice per group). Slices were incubated at 32.5 °C in artificial cerebrospinal fluid (aCSF; 126 mM NaCl, 2.5 mM KCl, 26 mM NaHCO_3_, 2.5 mM CaCl_2_, 1.5 mM MgCl_2_, 1.25 mM NaH_2_PO_4_, and 10 mM glucose) and pre-bubbled with 95% O_2_ and 5% CO_2_ for 1 h after the cutting procedure. At 1 h, tissue sections were transferred to an incubation chamber containing either normal aCSF solution or aCSF with 10μm Zn (zinc chloride; Sigma). Following a 15 min incubation period, the hippocampus was rapidly dissected, frozen, and stored in microcentrifuge tubes at −80 °C until they were analysed.

Protein extractions were performed, and total protein concentration was ascertained using a bicinchoninic acid (BCA) quantification kit (Thermo Fisher Scientific, Eugene, OR, USA). mBDNF levels were determined using an ELISA kit (Promega, Madison, WI, USA) according to the manufacturer’s instructions, with the exception of the acid treatment, which was not performed in order to measure mBDNF. 200 μg protein samples were run in duplicate and compared to a BDNF standard curve, run in a two-fold dilution series, to determine mBDNF levels. Based on the manufacturer’s protocol and a comparison paper of commercial BDNF ELISA kits, the intra-assay variation is 2–9%, and the inter-assay variation is 9–16% [46]. For the slice experiments, as slices from WT and ZnT3 KO mice were run on separate ELISA plates, the effects of zinc on the mBDNF protein were normalised to their respective genotypic baselines.

### 2.8. Statistical Analysis

Data obtained from the immunohistochemistry, BDNF, and Spatial Object Recognition experiments were analysed using a two-way factorial analysis of variance (ANOVA) with genotype (WT or ZnT3 KO) and type of housing (SH or EE) set as fixed factors (IBM SPSS Statistics, Version 19; Armonk, NY, USA: IBM^®^). A repeated measures ANOVA was used to analyse the data from the Morris Water Task. A post hoc Tukey’s test was used to determine differences where significant interactions were found. All ANOVA results are reported in Appendix A. To analyse the BDNF levels of hippocampal slices, an independent samples *t*-test was used. Tests for homogeneity of variances (Levene’s test for equality of variances) and sphericity (Mauchly’s test) are reported in the Appendix A. For all statistical tests, a *p*-value of <0.05 was considered statistically significant. All values are expressed as the mean ± standard error of the mean (Appendix A). Figures were created using GraphPad Prism Software (Version 9; San Diego, CA, USA).

## 3. Results

### 3.1. Knockout of ZnT3 Prevents Enriched Environment-Induced Proliferation without Affecting Baseline Proliferation

To examine whether vesicular zinc is involved in modulating adult hippocampal neurogenesis in an experience-dependent manner, we quantified the number of BrdU-positive cells within the granule cell layer and subgranular zone across the entire dentate gyrus in adult WT and ZnT3 KO mice exposed to either SH or EE. There was also a significant interaction between genotype and housing condition [*F*(1, 20) = 12.31, *p* = 0.002] (Appendix A). Post hoc Tukey’s tests revealed that, as expected, EE exposure significantly enhanced hippocampal cell proliferation in adult WT mice [*t*(20) = 5.23, *p* < 0.001]. In contrast, ZnT3 KO mice showed no increase in neurogenesis following EE exposure [*t*(20) = 0.25, *p* = 0.994] (Figure 1A,A’). Additionally, while baseline proliferation in the SH condition was not significantly different between the WT and ZnT3 KO mice [*t*(20) = 0.02, *p* > 0.999], there was a significant difference between the WT-EE and ZnT3 KO-EE mice [*t*(20) = 4.98, *p* < 0.001] (Figure 1B). Therefore, while knockout of the ZnT3 gene did not affect baseline levels of proliferation, it did abolish EE-induced increases in neurogenesis.

### 3.2. Vesicular Zinc Is Necessary for Enriched-Environment-Induced Survival of New Neurons

In addition to the number of proliferating cells, the number of surviving cells in the granule cell layer and subgranular zone of the hippocampal dentate gyrus was assessed. Brains were collected three weeks after the final BrdU injection, and the number of surviving cells was assessed by counting BrdU-positive cells. Our statistical analysis revealed a significant interaction between genotype and housing condition [*F*(1, 19) = 17.71, *p* < 0.001] (Appendix A). Post hoc Tukey’s tests showed that the KO-EE mice demonstrated significantly decreased levels of cell survival relative to WT-EE mice [*t*(19) = 6.19, *p* < 0.001] (Figure 2A,A’). Cell survival in WT and ZnT3 KO mice in the SH condition was not significantly different [*t*(19) = 0.39, *p* = 0.979]. In addition, cell survival in ZnT3 KO-EE mice was not different from that in either the WT- or ZnT3 KO-SH group [*t*(19) = 0.57, *p* = 0.940; *t*(19) = 0.94, *p* = 0.783, respectively] (Figure 2B). These results indicate that vesicular zinc is also necessary for EE-dependent increases in the survival of new cells.

Given the significant decrease in cell survival in the ZnT3 KO-EE group, we next examined the neuronal phenotype of the surviving cells (Figure 2C–C”)). There was no significant interaction between genotype and housing condition [*F*(1, 19) = 0.29, *p* = 0.596]. While there was no significant effect of genotype for neuronal phenotype [*F*(1, 19) = 0.01, *p* = 0.910], there was a significant effect of housing condition [*F*(1, 19) = 92.45, *p* < 0.001] (Appendix A). Despite the difference in the numbers of cells surviving between mice in the WT-EE and the other three groups (Figure 2B), the percentage of BrdU-positive cells co-labelled with NeuN was significantly elevated by EE in both WT and ZnT3 KO mice compared to SH groups (Figure 2D).

The differences in the number of surviving cells between WT and ZnT3 KO mice exposed to EE could be due to differences in the levels of cell death. For instance, we may have observed greater numbers of surviving cells in WT-EE mice because there is less cell death occurring. Thus, as a measure of cell death, we quantified the number of pyknotic nuclei (Figure 3A). Our quantification of pyknotic nuclei revealed a significant interaction between genotype and housing condition [*F*(1, 19) = 30.94, *p* < 0.001] (Appendix A). This interaction was due to significantly fewer numbers of pyknotic nuclei counted in the WT-EE mice compared to the ZnT3 KO-EE mice [*t*(19) = −8.68, *p* < 0.001]. No significant difference between WT and ZnT3 KO mice was found in the SH conditions [*t*(19) = 1.04, *p* = 0.729] (Figure 3B). These results indicate that EE significantly decreased cell death in WT mice but had no effect on ZnT3 KO mice.

Since changes in hippocampal cell survival are correlated with changes in granule cell layer volume [47], we next examined whether there were any differences in granule cell layer volume. Our analysis revealed a significant interaction between genotype and housing condition [*F*(1, 19) = 19.72, *p* < 0.001] (Appendix A). Post-hoc Tukey’s tests showed that this interaction was due to a significantly smaller hippocampal volume in the ZnT3 KO-EE group relative to the WT-EE group [*t*(19) = −6.36, *p* < 0.001]. There was no significant difference in hippocampal volumes between genotypes in the SH condition [*t*(19) = 0.24, *p* = 0.995] (Figure 3C).

### 3.3. Mice Lacking Vesicular Zinc Show No Improvement in Spatial Memory following Environmental Enrichment

Given that EE enhances hippocampal neurogenesis and typically improves spatial memory [18], we next utilised two behavioural assays, the Spatial Object Recognition and Morris Water tasks, to examine hippocampal-dependent spatial memory. Analysis of novel location investigation ratios in the Spatial Object Recognition task revealed a significant interaction between genotype and housing condition [*F*(1, 35) = 7.12, *p* = 0.011] (Appendix A). WT-EE mice showed a significant improvement in spatial memory compared to WT-SH [*t*(35) = 3.92, *p* = 0.002], ZnT3 KO-SH [*t*(35) = 3.90, *p* = 0.002], and ZnT3 KO-EE [*t*(35) = 3.70, *p* = 0.004] mice, which were not significantly different from one another (Figure 4A).

Hippocampal-dependent spatial learning and memory were also assessed with the Morris Water Task. A repeated-measures ANOVA found no significant effect of genotype [*F*(1, 35) = 0.12, *p* = 0.732] or housing condition [*F*(1, 35) = 0.18, *p* = 0.678] on swim speed. As such, escape latency to the hidden platform was used to measure performance. In our analysis of escape latency, we found a significant interaction between genotype and housing condition [*F*(1, 35) = 12.18, *p* = 0.001]. A post hoc Tukey’s test revealed that the WT-EE mice had significantly faster escape latencies than mice in all other groups [WT-SH: *t*(35) = 4.18, *p* = 0.001; ZnT3 KO-SH: *t*(35) = 2.73, *p* = 0.018; ZnT3 KO-EE: *t*(35) = 3.42, *p* = 0.002]. There was no significant difference amongst any of the other groups. As expected, there was a mean effect of trial day [*F*(2, 70) = 29.16, *p* < 0.001], indicating that all groups improved over the three days of testing (Figure 4B). Overall, these results indicate that normal levels of vesicular zinc are necessary for the enhancement of spatial memory following EE.

### 3.4. Zinc, Environmental Enrichment, and BDNF

BDNF is an important mediator of structural plasticity and neuronal viability and has been shown to be necessary for EE-induced neurogenesis [22]. Thus, we measured mature BDNF levels in the hippocampi of WT and ZnT3 KO mice under SH and EE conditions. Our analysis of BDNF levels in the hippocampus revealed a significant interaction between genotype and housing condition [*F*(1, 28) = 5.28, *p* = 0.029]. WT mice showed a significant elevation in BDNF levels with EE housing compared to SH housing [*t*(28) = 3.77, *p* = 0.004], while BDNF levels in the hippocampus of KO-EE mice were not significantly different from those measured in SH animals, regardless of genotype (Figure 5A).

To examine if ZnT3 KO mice were able to increase BDNF levels in response to zinc stimulation, we bypassed the inability of vesicular zinc to be released by exposing hippocampal slices from WT and ZnT3 KO mice to 10 µM Zn. An independent samples *t*-test found that the application of zinc significantly increased levels of mature BDNF in the hippocampus of both WT and ZnT3 KO mice when compared to their respective control groups [WT-Zn vs. WT: *t*(14) = 2.33, *p* = 0.018; ZnT3 KO-Zn vs. ZnT3 KO: *t*(11) = 2.19, *p* = 0.025] (Figure 5B).

## 4. Discussion

The purpose of this study was to determine whether vesicular zinc plays a role in experience-dependent modulation of adult hippocampal neurogenesis and hippocampal-dependent behaviour. Our results indicate that vesicular zinc is necessary for EE-induced neurogenesis, increased hippocampal levels of mBDNF, and the cognitive improvements mediated by EE. Unlike controls, ZnT3 KO mice did not show EE-induced increases in cell proliferation, cell survival, neurogenesis, granule cell layer volume, or mBDNF levels. Furthermore, the level of cell death in ZnT3 KO mice was unaffected by EE, and they did not show any improvement in their performance in the Spatial Object Recognition task or the Morris Water task. As such, these data highlight a potentially important role for the contribution of vesicular zinc to mediate experience-dependent changes in plasticity and behaviour.

As an essential mineral, zinc is necessary for neurogenesis via its ability to regulate proliferation, survival, and apoptosis during development and adulthood [16]. Utilizing broad zinc manipulations such as dietary zinc deficiency and systemic zinc chelation, several studies have found that zinc is necessary for hippocampal neurogenesis during development and adulthood [48,49]. Although effective at reducing zinc levels, systemic zinc manipulations do not differentiate between vesicular zinc and zinc’s role as an essential mineral. Our results emphasise this difference, as the specific loss of vesicular zinc by genetic knockout of ZnT3 had no effect on baseline neurogenesis, while broad zinc manipulations such as dietary zinc deficiency and zinc chelation decrease neurogenesis [48,49]. Instead, we found that ZnT3 KO mice are only impaired in the upregulation of hippocampal neurogenesis, mBDNF levels, and hippocampal-dependent behaviour, following EE. Thus, our results suggest that EE-induced enhancement of these factors is vesicular zinc-dependent.

Our findings complement previous findings in which experience-dependent changes in the levels of vesicular zinc occur within primary somatosensory cortex following sensory manipulations [50,51,52]. The degree of change in vesicular zinc levels can also be enhanced by EE [53], and thus, it is possible that vesicular zinc levels in the hippocampus could also be regulated in a similar manner following EE exposure. This regulation could be vital for experience-dependent plasticity within the hippocampus, as many of the neuromodulatory effects of zinc are concentration-dependent [2].

The release of vesicular zinc could regulate the activity of numerous signaling pathways vital for increasing hippocampal neurogenesis during enrichment. Zinc plays a pivotal role in regulating several factors that are essential for proliferation, including thymidine kinase, cyclin accumulation, and PI3k/Akt and extracellular signal-regulated kinases (ERK1/2) signaling (reviewed in Rink [54]). Cell survival could also be enhanced by zinc through the activation of the pro-neurogenic neurotrophic factor, mBDNF [13]. BDNF signaling during enrichment is a vital component of both the increased proliferation of neural progenitor cells as well as their long-term survival [15,55]. The deficient hippocampal mBDNF levels in enriched ZnT3 KO mice could explain the lack of an increase in adult hippocampal neurogenesis and neuronal survival that we saw following EE, which might imply that vesicular zinc is a pre- or co-requisite for the pro-neurogenic effects of mBDNF.

Several reports have indeed shown a strong link between exogenous zinc application and mBDNF/TrkB signaling in cell culture. These studies demonstrated that zinc is capable of TrkB activation either by converting proBDNF to mBDNF via the activation of MMPs and tissue plasminogen activator (TPA) [13,56] or by directly transactivating TrkB independently of BDNF [7,57]. Thus, altering vesicular zinc levels could provide a potent mechanism to increase the conversion of proBDNF to mBDNF. While controversial, there is also experimental evidence that neuronal cells can store and secrete proBDNF into the extracellular space [58]. The co-release of vesicular zinc with proBDNF could facilitate the conversion of proBDNF to mBDNF through zincergic activation of MMP or TPA. Enrichment also robustly increases the activity of MMPs and TPA, resulting in increased conversion of proBDNF to mBDNF [59,60]. Zinc can also complex with the N-terminal domain of BDNF, improving its structural stability [61]. Therefore, vesicular zinc signaling might be a key regulator of BDNF signaling, not only by catalyzing the conversion of proBDNF to mBDNF but also by stabilizing the structure of mBDNF. Future studies could measure levels of proBDNF to examine if the proteolysis capacity to cleave proBDNF into mBDNF is enhanced by the presence of vesicular zinc. Additionally, confirming whether TrkB phosphorylation is enhanced would be a valuable undertaking. Alternatively, it is possible that vesicular zinc can also act upstream of BDNF gene expression. One way to examine this would be to assess BDNF mRNA expression in response to EE. Vesicular zinc release has also been shown to activate the mitogen-activated protein kinase (MAPK) pathway in the hippocampal mossy fibers, leading to downstream effects on targets such as cyclic AMP response element-binding protein (CREB), which can stimulate BDNF gene transcription [62]. Previous studies have also proposed that zinc can stimulate CREB to increase BDNF expression through increased ERK1/2 activation and IGF-1 expression [63,64].

Interestingly, we found that the volume of the granule cell layer was increased in WT mice exposed to EE but not in ZnT3 KO mice, suggesting that vesicular zinc may also be necessary for EE-induced increases in granule cell layer volume. Enrichment has previously been shown to increase granule cell layer volume, with one study reporting that environmental enrichment increased the granule cell layer volume by 15% [21]. Wheel running has also been shown to increase the volume of the granule cell layer [65,66].

Changes in factors such as BDNF and other genes and gene products associated with regulating plasticity can in turn influence plastic changes related to learning and memory, which may explain why ZnT3 KO mice did not show any improvements in spatial memory following EE. Though the mechanism through which vesicular zinc is involved in EE-dependent changes in hippocampal-dependent learning and memory remains unclear, our results suggest that vesicular zinc could be modulating memory through its effect on neurogenesis, or BDNF. As mentioned previously, subsequent studies are needed to elucidate what these mechanisms are and to delineate the individual contributions that altered neurogenesis and BDNF signaling have on the cognitive improvements induced by EE. Interestingly, the necessity of vesicular zinc signaling for hippocampal plasticity may also extend to plasticity induced by injury, as ZnT3 KO mice do not show increased hippocampal cell proliferation following hypoglycemic brain insult [49].

The present study has important limitations that need to be addressed in future studies. The first is that the current study investigated the whole dentate gyrus. However, functional differences have been found between the dorsal and ventral hippocampus, with the former suggested to be primarily involved in cognitive functions, while the latter has been implicated in regulating emotional and motivated behaviours [67]. Thus, it may be worthwhile for future studies to examine the differences between the dorsal and ventral hippocampus in the context of vesicular zinc and the functional implications of such differences. The developmental necessity of zinc also presents a potential confound for the current study, as the lack of ZnT3 during development could alter the development of the hippocampal circuitry in ZnT3 KO mice. However, three lines of evidence argue against this as a possibility. First, there is minimal expression of ZnT3 in the hippocampus until after postnatal day 7 [68]. Although the amount of ZnT3 protein begins to increase after this, the actual amount of vesicular zinc released on postnatal day 16 is still negligible, estimated to be only 5–10% of what is released in adults [69]. Second, in the present study, ZnT3 KO mice kept in SH conditions were indistinguishable from WT controls in regards to neurogenesis and hippocampal-dependent behaviours. While our results contrast previous studies in which infusions of zinc chelators were found to impair several hippocampal-dependent behaviours [70,71,72,73], these approaches are not without limitations either. Infusion of zinc chelators could alter basal concentrations of extracellular and intracellular zinc, likely causing more global differences in zinc and affecting numerous signaling pathways. Additionally, the use of zinc chelators as a tool for studying vesicular zinc is further limited by the efficacy of extracellular zinc chelators, many of which cannot chelate zinc rapidly and are therefore, unable to block fast zinc signals that likely occur following vesicular zinc release [74,75,76]. They also chelate calcium and magnesium, rendering them ineffective in studying the specific role of vesicular zinc in synaptic signaling [77]. Furthermore, several other studies have also found no significant deficits in the hippocampal-dependent behaviour of young-adult ZnT3 KO mice raised in SH conditions [11,12]. Third, hippocampal slices from adult ZnT3 KO mice showed elevations in mBDNF levels in response to exposure to exogenous zinc, demonstrating that the system is still intact and responsive to zinc stimulation. Taken together, this evidence suggests that vesicular zinc signaling is not important for normative adult neurogenesis, mBDNF levels, or hippocampal-dependent behaviour. Instead, these data suggest that vesicular zinc is an essential regulator of experience-based changes in adult hippocampal neurogenesis under conditions of enrichment, but the exact mechanisms by which vesicular zinc does this remain to be elucidated. Ultimately, the results from this study present a novel perspective on mediators of experience-dependent plasticity within the hippocampus and indicate the potential impact of vesicular zinc signaling as a prime target for modulating plasticity, which may serve in the development of novel treatments in neurological conditions where mechanisms of neuronal plasticity are impaired.

## Figures and Tables

**Figure 1 cells-12-00883-f001:**
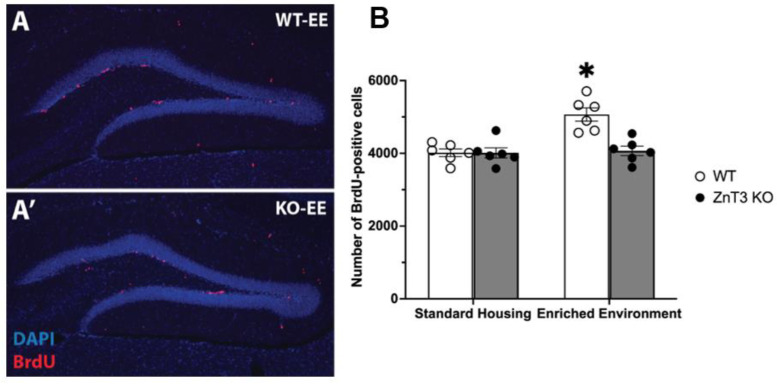
Cell proliferation in the hippocampus of WT and ZnT3 KO mice exposed to SH or EE (*n* = 24). Representative photomicrographs of DAPI (blue) and BrdU-immunopositive cells (red) in the subgranular zone of a (**A**) WT and (**A’**) ZnT3 KO mouse housed in EE. (**B**) The levels of cell proliferation seen in WT and ZnT3 KO mice placed in the SH condition did not differ. However, while WT mice in EE showed a significant increase in cell proliferation, the level of proliferation in ZnT3 KO-EE mice was significantly lower and was no different from levels in WT and ZnT3 KO mice in the SH condition. Error bars represent ± standard error of the mean. * indicates *p* < 0.001.

**Figure 2 cells-12-00883-f002:**
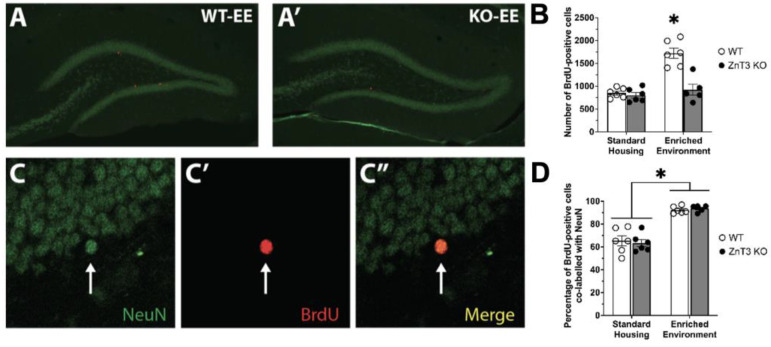
Cell survival in the hippocampus of WT and ZnT3 KO mice exposed to SH or EE (*n* = 23). Representative photomicrographs of BrdU-immunopositive cells (red) in the granule cell layer of a (**A**) WT and (**A’**) ZnT3 KO mouse housed in EE. (**B**) EE exposure significantly increased the survival of proliferating cells in WT mice but had no effect on ZnT3 KO mice. (**C**) Confocal microscopic images showing cells expressing the mature neuronal marker NeuN (green), with a BrdU-positive cell in the same plane ((**C’**), red). (**C”**) The channels are merged to show co-localisation. (**D**) Exposure to EE increased the percentage of BrdU-positive cells co-expressing NeuN in both WT and ZnT3 KO mice. Error bars represent ± standard error of the mean. * indicates *p* < 0.001.

**Figure 3 cells-12-00883-f003:**
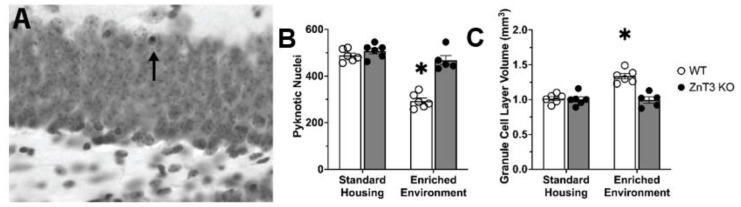
Cell death and granule cell layer volume in WT and ZnT3 KO mice exposed to SH or EE (*n* = 23). (**A**) A representative photomicrograph of cresyl violet-stained sections of the granule cell layer reveals pyknotic nuclei that are characteristic of dying cells (arrow). (**B**) The level of cell death did not differ between WT and ZnT3 KO mice in SH. EE significantly decreased cell death in WT mice but had no effect in ZnT3 KO mice. (**C**) There were no differences in granule cell layer volume between WT and ZnT3 KO mice in the SH condition. Granule cell layer volume is significantly increased in WT but not in ZnT3 KO mice exposed to EE. Error bars represent ± standard error of the mean. * indicates *p* < 0.001.

**Figure 4 cells-12-00883-f004:**
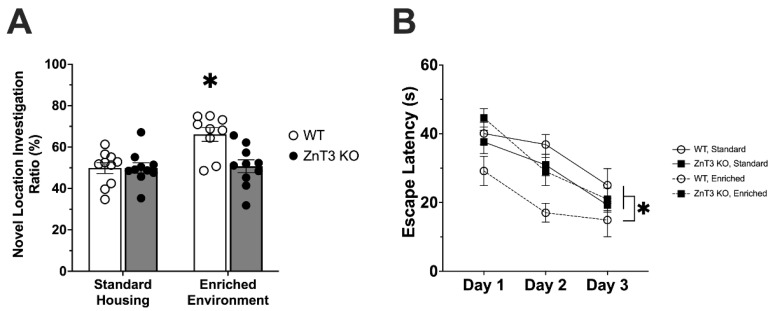
Spatial learning and memory in WT and ZnT3 KO mice exposed to SH or EE (*n* = 39). (**A**) EE exposure increases the investigation ratio of a novel object for WT mice, but not for ZnT3 KO mice, in the spatial object recognition task. (**B**) Escape latencies in the Morris Water task across the three testing days reveal that the performance of WT mice in the EE housing condition is significantly better than ZnT3 KO-EE, as well as that of WT or ZnT3 KO mice in SH. Error bars represent ± standard error of the mean. * *p* < 0.010.

**Figure 5 cells-12-00883-f005:**
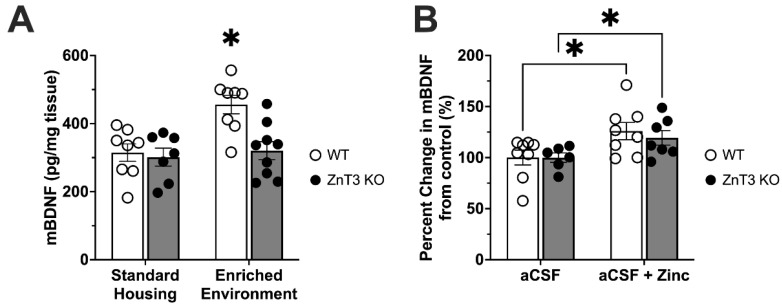
Measurements of mature BDNF (mBDNF) in the hippocampus of WT and ZnT3 KO mice. (**A**) The amounts of mBDNF measured in the hippocampus of WT and ZnT3 KO mice in SH did not differ. EE significantly increased mBDNF in the hippocampus of WT mice but not in ZnT3 KO mice (*n* = 32). (**B**) Effect of bath application of 10µM ZnCl on mBDNF levels in the hippocampus of WT and ZnT3 KO mice. Exogenously applied Zn significantly increased the level of mBDNF in the hippocampus of both WT and ZnT3 KO mice (*n* = 29). Error bars represent ± standard error of the mean. * indicates *p* < 0.050.

## Data Availability

The data presented in this study are available on request from the corresponding author upon reasonable request.

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
