# Peer review of "Environmental Enrichment Engages Vesicular Zinc Signaling to Enhance Hippocampal Neurogenesis"

_cells, 2023, doi:10.3390/cells12060883_

Round 1

Reviewer 1 Report (Previous Reviewer 2)

I would like to thank the authors for addressing my suggestions so thoroughly

Author Response

Reviewer 2 Report (New Reviewer)

Review of: “Environmental Enrichment Engages Vesicular Zinc Signaling 2
to Enhance Hippocampal Neurogenesis” submitted for publication in Cells

This paper describes a set of experiments designed to determine the possible cognitive function of vesicular zinc. Wild type mice were compared to mice with a genetic knockout of a zinc transporter gene. Enriched housing was used as a second variable, known to enhance hippocampal function. Cell proliferation, adult neurogenesis, pyknosis, and BDNF levels were assessed in the hippocampus as indices of neuroplasticity, and the Morris water maze and object location task were used to assess spatial memory. The experiment is well designed, and the paper is well written overall. Some of the methods could be explained more clearly and I am concerned about interpretation of some of the histological results. My major comments are itemized below, with minor grammatical and formatting issues listed subsequently.

Major comments
1. In the Introduction, it is mentioned that the ZnT3 KO mice have reduced levels of proBDNF (line 46), but zinc also promotes the conversion of proBDNF to mBDNF (line 53). This apparent contradiction should be clarified either in the Introduction or the Discussion.

2. It is also stated in the Introduction that adult neurogenesis “plays and important role in hippocampal-dependent learning and memory” (lines 60-61). A 13-year-old review article is cited to support this statement. The function of adult neurogenesis is a complex and controversial topic. Perhaps this complexity could be acknowledged, and the authors could cite some sources that specifically demonstrate the relationship between adult neurogenesis and water maze or object recognition tasks? The same problem occurs in the Discussion, where a positive relationship between neurogenesis and spatial memory seems to be assumed. Along these same lines, the cognitive function of cell proliferation within the adult brain is even more ambiguous than neuron survival levels. Why both of these measures were quantified was never addressed in the Introduction or Discussion.

3. A few more details would be useful regarding housing conditions. What type of diet was used specifically? Were animals housed with siblings? Why were only males used for the experiments?

4. Many different BrdU injection protocols are used. It would be helpful to cite a source that indicates that the researchers’ protocol does not have toxic effects. Three injections of 100 mg/kg separated by only 6 h seems like a relatively high does. Also, why was a 3-week period chosen for the cell survival assay?

5. It is stated that tissue was “collected as 6 adjacent series” (line104) The term “adjacent” is confusing here. What was the distance between sections in each series that was used for histological labeling?

6. Some aspects of the BrdU-labeling procedure seem to be missing (lines 105-115). Was the HCl neutralized with a base? Was a blocking step used? Was the tissue permeabilized with Triton-X or another surfactant? What type of mounting media was used? Regarding cresyl violet staining, it would be useful to either provide a citation or a few more details regarding the procedure.

7. It would be useful to explain more clearly how colocalization of BrdU and NeuN was assessed. Were cells simply visualized throughout the Z-stack? Was some threshold above background staining required for a positive NeuN label?

8. Presumably, pyknotic cells were counted throughout the GCL + SGZ, but this should be clarified (lines 131-134).

9. It is stated that mice underwent behavioral testing “three weeks after SH or EE exposure” (line 140). Did mice remain in their respective housing conditions during the period of behavioral testing?

10. Regarding water maze testing, what was the size of the pool? What was the water temperature? What was the length of the inter-trial interval within days? What type of tracking software was used?

11. For the BDNF kits, the authors should confirm that the ELISA kits are indeed measuring exclusively mBDNF rather than total BDNF. What was the cross-reactivity level with proBDNF? It would also be useful to provide intra-assay and inter-assay coefficients of variation for the ELISA kits.

12. An investigation ratio is used to describe the object recognition data, but this does not seem to be defined in the Methods. This seems important, as a number of different methods have been used by past researchers to assess this task.

13. There is some inconsistency in the reporting of the statistical results. Just reporting the interaction effect followed by pairwise post-hoc analyses seems reasonable. However, for the proliferation data it is stated that “as expected, EE 205 exposure significantly enhanced hippocampal cell proliferation in adult WT mice [F(1, 20) 206 = 27.41, p < .001]” and that “ZnT3 KO mice showed no increase in neurogenesis following 207 EE exposure [F(1, 20) = 0.08, p = .787].” This is confusing, as the description seems to be for post-hoc analyses, but the associated statistics seem to apply to the main effects of housing and genotype. Regarding the water maze data, were there no significant interaction effects involving day? For the slice data (lines 323-329), why were results reported only as t-tests (Fig. 5B) rather than using 2-way ANOVA as was used for all other analyses?

14. Perhaps one of the most unexpected results was the enrichment-induced increase in GCL volume among the wildtype mice (Fig. 3C). Are there any past examples of environmental enrichment actually increasing the size of the GCL? This result is barely mentioned in the Discussion. Is it possible that this was just a spurious effect due to the small sample size? Give this variation in volume, it would seem prudent to analyze the other histological variables (BrdU and pyknosis) as cell densities rather than just total count estimates. If there are no differences in cell densities, the results seem more difficult to interpret. Perhaps increased neurogenesis is simply due to an increased size of the GCL in the WT EE group?

15. The mBDNF data are reported as pg/mg tissue (Fig. 5A). Was this tissue wet mass? Or was the estimate of total protein concentration incorporated into these values in some way? This is somewhat unclear in the Methods section.

16. The first paragraph of the Discussion summarizes all the key findings. It would seem important to mention the cell proliferation, pyknosis, and GCL volume data at least briefly in this paragraph.

17. The Discussion paragraph about limitations (lines 418-429) could be deleted entirely. It is mostly redundant and quite wordy.

Minor comments
1. Line 35 does not need a comma: “signaling pathways, by binding” should be “signaling pathways by binding”.

2. Assuming that Cells uses SI units, all units of time (seconds, minutes, hours) should be abbreviated throughout (s, min, h).

3. Fig. 5 has graphs in a different format that all of the other graphs, with the x-axis labels and the legend flipped. I would recommend keeping all of the graphs in a consistent format.

4. There are tracked changes in red text throughout the Discussion section, which seems sloppy. It is unclear whether this was a mistake of the authors or the editors, but all tracked changes should be eliminated for the final draft.

5. Line 373 states that “Cell survival could also be enhanced 372 by zinc through the activation of the pro-neurogenic tyrosine kinase, mBDNF [13].” The end of this sentence needs to be reworded, as it implies that mBDNF itself is a tyrosine kinase.

6. Line 439: “While these results contrasts” should be “While our results contrast”.

7. Lines 444-449 is a run-on sentence that should be broken up.

Author Response

Reviewer 3 Report (New Reviewer)

The article by Chrusch et al describes the response to environmental enrichment on hippocampal neurogenesis in wild type and ZnT3 KO mice. The results are interesting, as they show that KO mice are unable to respond to EE with an enhancement of neurogenesis, and display no improvements in different behavioral tests. The study is sound, a few remarks can be made before publication:

·         Only male mice are used and one can wonder about the female, since neurogenesis is gender dependent, and female Zn T3 KO mice display different abnormalities compared to male.

·         The authors show that pyknotic cells are more abundant in KO mice, which explain why there is no improvement in response to EE. Can cresyl violet be used to measure cell death? If so, the reference used should be modified, because the one cited (22, by Huo et al) does not use it but only TUNEL and active caspase 3. Very simply, DAPI or Hoechst can be used to visualize fragmented nuclei. Figure 1 shows dorsal DG and the authors state that the whole DG has been inspected. Does the ventral Hp provide similar findings?

·         The lines 256 to 270 are not very clear, especially the first sentence, because of the repetitions of double negative, and should be clarified, as it is an important point in the article.

Author Response

This manuscript is a resubmission of an earlier submission. The following is a list of the peer review reports and author responses from that submission.

Round 1

Reviewer 1 Report

Authors found that BrdU-labeled proliferating cells and survived cells are decreased in ZnT3KO only under EE, not SH. In addition to the histological analysis, behavioral analyses like MWM task is impaired in the ZnT3KO only under EE. They also found that this inhibition might be related to the decreased production of mature mBDNF.

The story seems acceptable; however, I have several questions to confirm the signaling mechanism related to decreased mature BDNF, as the authors discussed.

#1  Could authors measure proBDNF to conclude that proteolysis is decreased by decreased zinc?

#2 Could authors confirm if the TrkB phosphorylation activated by zinc is decreased in the DG or not? 

#3 Could authors measure BDNF transcript level in the DG?

These results facilitate the entire story of this paper.

 I think ZnT3 expression analysis is critical. According to Allen brain atlas, ZnT3 expression is high in the dentate gyrus and hippocampus.

#4 Could authors confirm ZnT3 protein is expressed in the neural stem cell or differentiated neuron by immunohistochemistry?

Reviewer 2 Report

The authors of this manuscript present a very well designed, comprehensive, and compelling analysis of the modulatory role of vesicular zinc on hippocampal neurogenesis.

I have only minor comments to the manuscript (in order of appearance)

Line 136: Reference 23 is a poor reference for Cavalieri estimator. Please refer either to Gundersen’s original review (10.1111/j.1699-0463.1988.tb05320.x), or the text of Howard and Reed (Howard, C. V. and M. G. Reed (2010). Unbiased Stereology. Three-dimensional Measurements in Microscopy. Liverpool, QTP Publications), or the recent review of Slomianka (10.1002/cne.24976). 

Line 163: this paragraph might benefit if the zinc slice incubation could be mentioned in the title

Paragraph starting line 256 and Figure 3: The graph B in Figure 3 implies that it is not an increase in cell death in KO EE mice, but a failure to decrease cell death in response to an EE. An increase vs a failure to decrease may be mechanistically quite different, and some precision in the description is therefore not just semantics.

Paragraph starting line 412: I agree that the lack of zinc to effect behavioral performance under SH conditions is a consistent observation in ZnT3 KO mice. This seems to contrast the reasonably consistent effect of the acute local or systemic modulation of synaptic zinc levels on a variety of hippocampus dependent behaviors (e.g., 10.1006/nlme.1999.3931, 10.1016/j.bbr.2004.03.003, 10.1038/s41598-018-28083-9). Is it possible that the lack of KO effects under SH conditions may reflect some developmental compensation? The authors may choose (!) to include a short contrast of KO vs acute effects if they think it will further their case and not distract from the main message of the manuscript.

Round 2

Reviewer 1 Report

The authors did not do any experiments for revision as I suggested.
I believe they should have done the expression analysis using wild type mouse separately.